# Eco-Friendly Extraction to Enhance Antioxidants and Nutritional Value in *Arthrospira platensis*

**DOI:** 10.3390/foods14091510

**Published:** 2025-04-26

**Authors:** Massimo Milia, Viviana Pasquini, Piero Addis, Alberto Angioni

**Affiliations:** Food Toxicology Unit, Department of Life and Environmental Science, Campus of Monserrato, University of Cagliari, SS 554, 09042 Cagliari, Italy; viviana.pasquini@unica.it (V.P.); addisp@unica.it (P.A.); aangioni@unica.it (A.A.)

**Keywords:** *Arthrospira platensis*, biochemical compounds, ultrasound-assisted extraction, antioxidant activity

## Abstract

The cyanobacterium *Arthrospira platensis* (Spirulina) has a global annual production of approximately 18,000 tons. Spirulina has notable nutritional benefits and is a key component of dietary supplements. However, efficiently extracting its bioactive compounds poses challenges. This study aimed to develop an eco-innovative method to enhance Spirulina’s antioxidant and nutritional values. The extraction protocol included a maceration step in phosphate-buffered saline (PBS, pH 7.4) at 5 °C for 48 h, followed by ultrasound-assisted extraction (UAE) at 400 W and 30 kHz, with a frequency of 30 cycles per min (consisting of 1 s on/off cycles, for a total of 6 extraction cycles). The proposed methodology allowed for the quantitative recovery of high-value compounds from Spirulina raw material (control), with increased yields of total lipids (+20.29%), total fatty acids (+60.48%), allophycocyanin (Apc, +41.41%), phycoerythrin (Pe, +81.42%), carotenoids (+30.84%), and polyphenols (+65.99%), leading to a boost in antioxidant activity (+42.95%). Conversely, the recoveries of proteins (−16.65%), carbohydrates (−18.84%), and phycocyanin (Pc, −0.77%) were incomplete. This study suggests a green extraction approach using PBS coupled with UAE, with promising energy and cost savings and potential applications in the dietary supplement sector.

## 1. Introduction

Spirulina (Cyanophyceae; *Arthrospira* spp.) represents the most widely cultivated cyanobacteria, with over 18,000 tons of biomass produced annually, dominating the current microalgae market. The global Spirulina market size reached USD 577.0 million in 2023. The International Market Analysis Research and Consulting Group (IMARC) expects the market to reach USD 1287.8 million by 2032, exhibiting a growth rate (CAGR) of 9.33% during the period of 2024–2032 [1,2]. Spirulina is a fast-growing, adaptable, and protein-rich superfood [3] containing around 60–70% of protein by dry weight [4,5,6]. In addition, it has bioactive components with a high therapeutic potential, such as polyphenols, carotenoids, phycobiliproteins (PBPs), carbohydrates, vitamins, sterols, polyunsaturated fatty acids (PUFAs), and minerals [7,8]. Accordingly, most of the world’s Spirulina production is intended for dietary supplements, especially in sports nutritional products, to maintain, restore, or enhance energy levels and immune function, reducing oxidative stress and improving general well-being [9,10,11]. Dietary supplements containing Spirulina raw material or extracts are marketed as pills, tablets, capsules, liquids, and powders [12]. Spirulina-based products are characterized by a protective effect against muscle damage related to overtraining; they reduce blood lactate dehydrogenase, serum glucose, triglyceride, and cholesterol levels [13].

Spirulina’s amino acid composition fulfills FAO nutrition requisites [14]; however, the protein digestibility of microalgae used as single-cell protein represents a key limiting factor in humans and animals, decreasing the availability of essential amino acids [3,15,16,17]. Devi et al. evaluated the amino acid digestibility of Spirulina in humans, implementing a dual-isotope method. The authors found an 85.2% mean amino acid digestibility, with values ranging from 77.5% (lysine) to 95.3% (phenylalanine) [18]. A relatively low algae amino acid digestibility has been related to the cell walls, macronutrient composition, enzyme specificity, anti-nutritional factors, fiber, and absorptive capacities within the gastrointestinal tract [19,20,21,22]. 

Thus, the hydrolysis of covalent bonds in the cell wall matrix is recommended to facilitate the extraction of proteins and other biochemical compounds for their use in food and supplement applications, enhancing protein digestibility and solubility [23,24,25,26,27] through extracting phenolics [28,29], chlorophyll a (Chl a) [30], PBPs [31,32,33], lipids, and carotenoids [34,35,36]. Target compound recoveries are highly dependent on the extraction conditions applied. Conventional extraction depends on mechanical and chemical processes, with high production costs, long extraction times, and elevated temperatures. A prolonged time and high temperature can lead to the degradation of bioactive compounds, reducing the extracts’ overall bioactivity [37,38,39,40]. These approaches often result in the co-extraction of unwanted materials, a loss of selectivity, and residual solvents hazardous to human health and the environment [41,42]. These limitations require efficient and environmentally friendly extraction technologies to save time and energy, using green solvents in line with ecologically sustainable strategies [43,44]. In recent years, microwave-assisted extraction (MAE) and ultrasound-assisted extraction (UAE) have shown a promising capacity in terms of both extraction efficiency and eco-sustainability [45,46,47,48]. The cavitation movement imparted to the solution by UAE leads to the breaking of hydrophobic and hydrogen bonds, creating a series of microscopic channels with a sponge effect in the tissue, improving the solubility of proteins, allowing compounds of interest to be released more efficiently [25,26,27], maintaining integrity, and reducing the interference of impurities [46,49]. UAE has shown a high efficiency, reduced solvent consumption, moderate extraction times and costs, and ease of management. Furthermore, it allows for faster and more scalable production from the laboratory to industry [50].

The UAE technique, coupled with maceration as pre-treatment, has shown an increased extraction of bioactive compounds with high antioxidant activity [51,52].

However, some disadvantages should be considered when scaling up this technology to an industrial level. UAE can have a low uniformity when big samples are processed, since ultrasound waves decrease in intensity with distance from the emitter; moreover, pasty matrices reduce the penetration of the waves, which requires technical adjustment for proper action. Scaling up the system results in higher energy consumption and faces problems in controlling equipment parameters; therefore, to effectively have a broader impact, coupling with clean energy, such as wind power conversion, solar energy conversion, and hydraulic conversion, is mandatory.

Extraction yield is influenced by the target compounds’ polarity, chemical stability, solvent, temperature, and time [51,53]. Among “green solvents”, PBS is economical, non-toxic, and poses no risk of harmful residues or environmental contamination. PBS, maintaining a stable pH in extraction environments, can prevent protein denaturation and lipid peroxidation and allow for obtaining high yields of polyphenols, phycobiliprotein, and photosynthetic pigments [13,51,54].

In this work, the macronutrient and bioactive compound compositions were investigated to evaluate the feasibility and efficacy of the UAE technique coupled with low-temperature PBS maceration in extracting biomolecules from *Artrospira platensis* to obtain an extract with potential applications in the dietary supplement sector.

## 2. Materials and Methods

### 2.1. Standards and Reagents

Methanol (MeOH), hydrochloric acid (HCl), sodium hydroxide (NaOH), and chloroform (CHCl_3_), ultra-residue solvents of analytical grade, were purchased from Merck (Darmstadt, Germany). Sulfuric acid (at 96% and 0.5 N), sodium hydroxide (NaOH) (32%, 0.5 N and 1 N), phenol, KCl, CaCl_2_, Na_2_CO_3_, CuSO_4_, Na and K tartrate, Na_2_SO_4_ anhydrous, Na_2_HPO_4_, NaH_2_PO_4_, NaCl, D-glucose, α-amylase, and 2, 2 diphenyl-1-picrylhydrazyl were reagent grade and were purchased from Sigma Aldrich (Chemie, Munich, Germany), as well as the internal standard (IS) tritridecanoin (TAF C13:0). 2-amino-2-hydroxymethyl-propane-1,3-diol (Tris) was analytic grade, purchased from BDH, UK (Copens Scientific, Petaling Jaya, Selangor, Malaysia). The PBS solution was prepared in the laboratory by weighing 1.09 g of Na_2_HPO_4_, 0.32 g of NaH_2_PO_4_, and 9 g of NaCl, dissolving in MilliQ water until a 1 L final volume was achieved, and adjusting the pH to 7.4 with HCl (0.1 M), reaching a final concentration of Na_2_HPO_4_ (8 mM), NaH_2_PO_4_ (3 mM), and NaCl (154 mM).

Double-deionized water with a conductivity of less than 18.2 MΩ was obtained with a Milli-Q system (Millipore, Bedford, MA, USA). The FAME Mix standard (37 Component) for GC-FID analysis was purchased from SUPLECO (595 North Harrison Road, Bellefonte, PA, 16823-0048 USA). Thiamine hydrochloride (B1) > 98.0%, nicotinamide (B3) > 99.0%, and pyridoxine hydrochloride (B6) > 98.0% were purchased from TCI Europe N.V. (Tokyo Chemical Industry, Zwijndrecht, Belgium), riboflavin (B2) 99.43% was purchased from TargetMol Chemicals Inc. (Boston, MA, USA), vitamin B12 ≥ 98% and L-ascorbic acid 99% were purchased from Indagoochem (Barcelona, Spain), and folic acid (B9) 97.0–102.0% was purchased from Glentham, Life Sciences Ltd. (Corsham, UK).

### 2.2. Raw Material and Processing

Spirulina microalgae grown in raceway ponds were concentrated and dried in the factory following the commercial protocol (LiveGreen, Società Agricola SRL, Arborea, Oristano, Sardinia, Italy).

The dried Spirulina was powdered with a porcelain mortar, and an exact amount was mixed with 300 mL of PBS (pH 7.4) at a ratio of 1:14 (*w*/*v*), according to Benedetti et al. [55] and Tavanandi et al. [31]. The suspension was mixed with a magnetic stirrer for 1 h, macerated for 48 h in the dark at 5 °C, and subjected to UAE using an ultra-sonic GM extractor 1 (GM Solution, Cagliari, Italy). The UAE operational conditions were 30 cycles/min at 100% power (400 W, 30 kHz; 1 s ON and 1 s OFF) for 6 extractions. Every 30 cycles, the solution was allowed to stand for 1 min. The entire process was carried out in an ice bath to avoid degradation of the target thermosensitive compounds. After centrifugation, the supernatant was collected, freeze-dried (LIO5P DIGITAL Cinquepascal srl, Trezzano sul Naviglio, Milan, Italy), and weighed. Biochemical analyses were carried out on the dried Spirulina raw material (control) and the freeze-dried biomass from UAE extraction (extract).

### 2.3. Biochemical Analysis

#### 2.3.1. Ashes

The control and extract samples (0.5 g) were carbonized at 525 °C in a porcelain crucible for 8 h. Ashes are expressed as g·100 g^−1^ DW.

#### 2.3.2. Total Proteins

Protein content was evaluated following the Kjeldahl method [56]. Control and extract samples (500 mg) were put into a Kjeldahl flask with 0.5 g of Na_2_SO_4_, 10 mg of CuSO_4_, and 20 mL of H_2_SO_4_ 96%. The samples were digested in a Speed-Digester K-436 BÜCHI (Labortechnik GmbH, Essen, Germany) at 400 °C until the solution became colorless. The Kjeldahl flasks were then removed, cooled, and introduced into a distillation system VAPODEST 300 (C. Gerhardt GmbH & Co. KG, Königswinter, Germany). In total, 100 mL of MilliQ water and 80 mL of NaOH (32%, *w*/*v*) were automatically added to the Kjeldahl flask. The distillate was recovered in a 250 mL receiving flask, with 10 mL of H_2_SO_4_ (0.5 N) and methyl red indicator (10 drops). Distillation lasted 4 min, and the obtained solution was titrated with NaOH (0.5 N) until it changed from red to light yellow. The total protein content is expressed as g·100 g^−1^ DW, as follows:% protein = ((a − b) × c × 100 × K)/g(1)

a: mL of H_2_SO_4_ 0.5 N added to the collection flask (10 mL).

b: mL of titrant used (NaOH 0.5 N).

c: conversion factor mL of H_2_SO_4_ 0.5 N in g of nitrogen (0.007).

K: general nitrogen–protein conversion factor (6.25).

g: grams of sample.

#### 2.3.3. Total Carbohydrates

Carbohydrate analysis was conducted according to Dubois et al. [57]. Control and extract samples (15 mg) were put into a glass tube with 5 mL of HCl (1 M), sonicated in a water bath (15 min), and extracted in a boiling water bath (1 h at 100 °C). In total, 100 μL of the extract was collected in a glass tube with 1 mL of phenol solution (5% in deionized water, *w*/*v*) and 5 mL of H_2_SO_4_ (95%). The mixture was mixed, left at room temperature for 30 min in the dark, and measured at 488 nm using a Cary 50 UV–Vis spectrometer (Varian Inc., Palo Alto, CA, USA).

Quantification was assessed by a 5-point calibration curve of D-glucose (20–100 mg L^−1^) with r^2^ ≥ 0.997. This analysis was carried out in triplicate, and total carbohydrates are expressed as D-glucose equivalents g·100 g^−1^ DW.

#### 2.3.4. Total Lipids

Total lipid quantification was conducted according to Chen et al. and Pasquini et al. [58,59]. After suspending the control and extract samples in PBS, saponification with a NaOH 1 N/MeOH solution (75/25), and centrifugation, the supernatant was transferred to a Falcon tube and mixed with CHCl_3_/MeOH (2/1) and a KCl solution 0.88% (*w*/*v*). In total, 1 mL of the organic phase was transferred to a 2 mL HPLC vial and evaporated under nitrogen, and the remaining fat residue was weighed. The analysis was performed in triplicate, and total lipids are expressed as g·100 g^−1^ DW.

#### 2.3.5. Neutral Detergent Fiber (NDF)

An FIWE Advance Automatic Fiber Analyzer, VELP Scientifica (Usmate Velate (MB), Italy) was used to assess the neutral detergent fiber (NDF) according to Van Soest et al. [60]. Control and extract samples (500 mg) with 0.5 g of Na_2_SO_3_ were placed in a P2 glass crucible and subjected to digestion (1.5 h) using a neutral surfactant mixture (6.81 g Na_2_B_4_O_7_ × 10H_2_O + 18.61 g EDTA + 30 g C_12_H_25_NaO_4_S + 10 mL triethylene glycol + 4.56 g Na_2_HPO_4_ in 1000 mL of double-deionized water) and α-amylase. The samples were dried in an oven for 4 h (100 °C) and mineralized for 5 h at 525 °C. The analysis was carried out in triplicate, and NDF% concentration is expressed as g·100 g^−1^ DW, as follows:% NDF = (W1 − W0) × 100/sample (g)(2)W1 = Crucible weight (g) + sample weight after dryingW0 = Crucible weight (g) + sample weight after mineralizationSample (g) = grams of sample weighted

#### 2.3.6. Chl a and Carotenoids

The concentration of Chl a and total carotenoids was determined according to Singh et al. [61]. Control and extract samples (20 mg) were placed in a 15 mL Falcon tube with MeOH (4 mL); after 2 min of agitation in a vortex, the Falcon was left in a thermostatic bath at 70 °C for 3 min. The tubes were cooled at ambient temperature and centrifuged at 3154× *g*, 10 °C for 5 min (Centrifuge 5810 R, Eppendorf AG 22,331 Hamburg, Germany) to precipitate cell debris. The bright green supernatant was analyzed at 470, 666, and 750 nm in a UV–Vis spectrometer (Cary 50, Varian Inc., Palo Alto, CA, USA). The Chl a and total carotenoid concentrations were calculated according to Chamizo et al. [62].Chl a mg L^−1^ = 12.9447 (A665 − A750)(3)Carot mg L^−1^ = (4.08 × (A470 − A750) − 0.0117 × ChlA mg L^−1^)(4)Chla a mg g^−1^ = (Chl a mg L^−1^ × Vol Solv(L)/g Sample) × Dil(5)Carot mg g^−1^ = (Carot mg L^−1^ × Vol Solv(L)/g Sample) × Dil(6)

Analyses were carried out in triplicate, and the amounts of Chl a and total carotenoids are expressed as mg·100 mg^−1^ DW.

#### 2.3.7. Phycocyanin (Pc), Allophycocyanin (Apc), and Phycoeritrin (Pe)

Control and extract samples (10 mg) were dissolved in CaCl_2_ (4 mL, 1%, w v^−1^) according to Dianursanti et al. [63]. The procedure comprised 3 freezing cycles (1 h at −80 °C each) followed by thawing with warm water. The concurrent effect of the thermal shock and CaCl_2_ accelerated cell lysis, allowing for the extraction of the water-soluble cellular components. Finally, the tubes were centrifuged at 3154× *g*, 10 °C (Centrifuge 5810 R, Eppendorf AG 22,331 Hamburg) for 10 min to precipitate unwanted cellular components. The bright blue supernatant was analyzed by a UV–Vis spectrometer at 565, 620, 652, and 750 nm (Cary 50, Varian Inc.). The Pc, Apc, and Pe concentrations were calculated according to Zavřel et al. [64].Pc mg mL^−1^ = (((ABS620 − ABS750) − 0.474 × (ABS652 − ABS750))/5.34(7)Pc% = (Pc mg mL^−1^ × Vol Solv(L) mg S^−1^) × 100 × DF(8)Apc mg mL^−1^ = ((ABS652 − ABS750) − 0.208 × (ABS620 − ABS750))/5.09(9)Apc% = (Apc mg L^−1^ × Vol Solv(L) mg S^−1^) × 100 × DF(10)Pe mg L^−1^ = ((ABS565 − ABS750) – 2.41 × (Pc mg mL^−1^) − 0.849 × (Apc mg mL^−1^))/9.62(11)Pe% = (Pe mg L^−1^ × Vol Solv(L) mg S^−1^) × 100 × DF(12)

S = sample; DF = dilution factor.

Analyses were carried out in triplicate, and the final concentrations of Pc, Apc, and Pe are expressed as mg·100 mg^−1^ DW.

#### 2.3.8. Total Polyphenols

Total polyphenols were assessed using the Folin–Ciocalteu reagent method, according to Manconi et al. [65]. In total, 10 mg of control or extract samples was placed in a 15 mL Falcon tube with 10 mL of methanol. The tubes were shaken for 1 min and were subjected to ultrasound-assisted extraction (UA E) at 100% power (400 W, 30 kHz; 1 s ON and 1 s OFF, 3 min). Finally, the samples were centrifuged at 3154× *g* at 10 °C for 15 min (Centrifuge 5810 R, Eppendorf AG 22,331 Hamburg). In total, 100 µL of the extract and 100 µL of the Folin–Ciocalteu reagent were put into a 15 mL Falcon tube and vortexed (1 min). The mixture and 400 µL of a sodium carbonate solution (20% *w*/*v*) were agitated and sheltered (60 min, 18 °C in the dark). The total amount of polyphenols was assessed at 765 nm using a UV–Vis spectrometer (Cary 50, Varian Inc., Palo Alto, CA, USA) by comparing the absorbance with a 7-point calibration curve of Gallic acid (2.5–200 µg·mL^−1^; r^2^ ≥ 0.997). Analyses were carried out in triplicate, and total polyphenols are expressed as mg·g^−1^ gallic acid eq DW.

#### 2.3.9. Fatty Acid Extraction and GC-MS Analysis

Fatty acids (FAs) were extracted according to Breuer et al. [66]. Briefly, 50 mg of control and extract samples was weighed in a 15 mL Falcon (F1) with a CHCl_3_/MeOH solution (4 mL, 4:5, *v*/*v*) containing tritridecanoin acid (IS at 50 mg·L^−1^). After mixing, 2.5 mL of NaCl solution (1 M in MilliQ water and 50 mM in 2-amino-2-hydroxymethyl-propane-1,3-diol-Tris) was added, and the pH was adjusted to 7.0 using a HCl solution. The sample was vortexed in the presence of glass beads (5 min), sonicated in ice (10 min), and centrifuged at 3154× *g* at 15°C (5 min). The CHCl_3_ layer was transferred to a 12 mL glass tube, whereas the residual sample was re-extracted with 1 mL of CHCl_3_ and finally assembled in the glass tube. The CHCl_3_ phases were brought to dryness under a nitrogen flow. The lipidic residue was subjected to the transesterification of fatty acids by adding 3 mL of a methanol solution at 5% (*v v*) sulfuric acid. The solution was placed on a thermostatic plate at 70 °C for 3 h, with a 30 s vortex every 30 min. After cooling, 3 mL of MilliQ water and 3 mL of hexane were added, and the tube was vortexed for 1 min and centrifuged at 3154× *g* at 15°C for 5 min. The hexane phase was transferred to a 2 mL amber vial and analyzed with an Agilent 8860 Gas Chromatograph (GC) (Agilent Technologies, Santa Clara, CA, USA) coupled with a 5977C GC/MSD Detector. The column was an HP-5 ms Ultra Inert (Agilent GC Column, 30 m × 0.25 mm, and 0.25 µm thickness). The injector temperature was 250 °C. He was the carrier gas at 1.5 mL·min^−1^ and the injection volume was 1 μL, with a split ratio of 1:2.

The oven temperature gradient was T = 0 50 °C (4 min), T = 26 160 °C (5 °C·min^−1^), T = 76 210 °C (1 °C·min^−1^), and T = 93.5 280 °C (4 °C·min^−1^). The transfer line temperature was 280 °C. Ions were generated at 70 eV with positive EI and recorded at 1.5 scans·s^−1^ from 50 to 550 m·z^−1^. The electromultiplier was set at 1162 V. The analysis was conducted in triplicate.

The content of each fatty acid (FA mg·g^−1^ DW) was determined using the following formula [66]:FA mg·g^−1^ = (IS mg·sample^−1^) × {(Area of individual FAME)/[(Area of C13:0 FAME(IS) × Rel.Resp.Factor individual FAME)]}/(biomass (g))(13)

The concentration of the IS mg·sample^−1^ was calculated with the following formula:IS mg·sample^−1^ = 0.004 L × [IS] × (3 × MW FA C13:0)/(MW TAF C13:0)(14)

[IS]: 50 mg·L^−1^; MW FA C13:0: 214.34; MW TAF C13:0: 684.08.

The relative response factor of individual FAME was calculated with the following formula:Rel. Resp. Factor individual FAME = (conc. C13:0 FAME in calibration sol./conc. individual FAME in calibration sol.) × (Area of individual FAME in calibration sol./Area of C13:0 FAME in calibration sol.)(15)

#### 2.3.10. DPPH Method

A 2, 2 diphenyl-1-picrylhydrazyl (DPPH) test was carried out as described by Manca et al. [67]. Total polyphenol control and extract samples were mixed with a DPPH methanolic solution (25 μM) at a ratio of 1:50. Samples were incubated for 30 min in the dark at room temperature. Analyses were performed at 517 nm in a UV–Vis spectrophotometer (Cary 50, Varian Inc.). Quantification was assessed by a 5-point calibration curve of Trolox (0.025–1 mM, r^2^ value ≥ 0.997).

The test was carried out in triplicate, and the antioxidant activity is expressed as mg⋅g^−1^ Trolox eq DW.

#### 2.3.11. Water-Soluble Vitamins

Water-soluble vitamin extraction and analysis were carried out according to Rakuša et al., with minor changes [68]. Control and extract samples (20 mg·mL^−1^ in 0.5% glacial acetic acid) were homogenized by vortex for 1 min, sonicated in an ice bath for 10 min, kept in the dark (10 min at 5 °C), and centrifuged at 3154× *g* (10 min at 15 °C). The solution was filtered with a 0.45 µm nylon filter just before injection.

An ultra-high-performance liquid chromatograph UHPLC Infinity II (Agilent Technologies, Santa Clara, CA, USA) was used, equipped with a diode array detector (DAD), on reversed-phase Infinity Lab Poroshell 120 EC-C18, 3.0 × 150 mm, 2.7 μm, 1000 bar, Agilent settled at 20 °C. The mobile phase included A: 25 mM NaH_2_PO_4_·H_2_O, and B: methanol, with the following gradient: T = 0 97% A, T = 9 85% A, T = 10 80% A, T = 20 75% A, T = 30 45% A, and post-run 10 min. The flow rate was 0.3 mL·min^−1^, and the injection volume was 10 μL. UV–vis detection was performed at 246 nm (B1), 290 nm (B2), 260 nm (B3), 210 nm (B6), 268 nm (B9), 362 nm (B12), and 244 nm (C). Vitamins were identified by comparing retention time and UV-vis spectra. Stock standard solutions of the single vitamins were prepared in dark volumetric flasks with concentrations of around 1000 mg·L^−1^ in 0.1 M EDTA (chelating agent, stabilizer) and stored at 5 °C. Vitamin B9 was prepared in NaOH 1 mM due to its low water solubility. A 5-point calibration curve (0.25–5 mg·L^−1^, r^2^ ≥ 0.997) was prepared by diluting the intermediate working standards solution at 100 mg·L^−1^ in 0.5% (*v*/*v*) glacial acetic acid before use. Glacial acetic acid was used to improve the solubility of vitamins, facilitate their separation from other components, and prevent the degradation of sensitive vitamins (B1 and B2).

#### 2.3.12. Statistical Analysis

The experiments were performed in triplicate, and data are reported as the mean ± RSD%. One-way analysis of variance (ANOVA) was performed using the complete set of data acquired as a source of variation. Before performing the ANOVA test, variance homogeneity was confirmed with Cochran’s test. The statistical analyses were performed using STATGRAPHICS PLUS 5.1 professional edition (Statistical Graphics Corp., Rockville, MD, USA).

## 3. Results

The biochemical analysis of the control and extract samples revealed significant differences in their macronutrient composition (Table 1). After the extraction process, the total macronutrient content was as follows: total carbohydrates at 12.63 ± 6.06, protein at 52.23 ± 0.50, lipids at 6.21 ± 5.61, fiber at 0.35 ± 8.70, and ash at 18.55 ± 0.64 g·100 g^−1^ (mean ± RSD% DW). During extraction, carbohydrates and protein were only partially recovered at 81.16% and 83.35%, respectively. Conversely, lipids showed an increase of 20.29%, whereas neutral detergent fiber (NDF) decreased by nearly 90% (Table 1). The ash fractions were significantly enriched, increasing by 120%.

Total carotenoid and Chl a showed a higher value in the extracts of almost 30% (Table 2). Apc (4.58 ± 5.16 mg·100 mg^−1^ ± RSD% DW) and Pe (1.20 ± 4.33 mg·100 mg^−1^ ± RSD% DW) were higher in the extract at +41% and +81% (*p* < 0.001), respectively. Meanwhile, the Pc concentration showed no significant difference (*p* > 0.05) between the control and the extract (Table 2).

The total polyphenols concentrations were 5.94 ± 2.05 and 9.86 ± 5.64 (mg·g^−1^ ± RSD% gallic ac. eq.) (*p* < 0.001) in the control and extract, respectively. In addition, the DPPH assay showed a higher value in the extract compared to the control, at 21.10 ± 4.58 vs. 14.76 ± 6.40 (mg·g^−1^ ± RSD% Trolox eq) (Table 2).

The vitamin B group (B1, B3, B2, B6, B9, and B12) and vitamin C were investigated in the control and extracts; however, only B3 was detected, with values of almost half in the extract (1.54 ± 3.10 mg·100 g^−1^ ± RSD%) compared to the control (3.49 ± 5.04 mg·100 g^−1^ ± RSD%) (Table 2).

Six compounds comprised 95% of the total fatty acids (FAs), exhibiting an increase of nearly 60% in the extract. The extraction procedure did not alter the FA profile; palmitic acid and linoleic acid were the most abundant, averaging 48.35% and 33.40%, respectively. Oleic acid and γ-linolenic acid accounted for 5.78% and 7.93%, followed by palmitoleic acid at 2.76%. Stearic acid accounted for only 1.78% of the total FAs.

When comparing the fatty acids in the control and the extract, palmitoleic acid, γ-linolenic acid, and linoleic acid exhibited the most significant differences. Their concentrations varied from 0.53 ± 10.89% to 1.01 ± 7.12%, from 1.62 ± 9.36% to 2.77 ± 7.11%, and from 6.90 ± 6.63% to 11.51 ± 8.29% (*p* < 0.01) (mg·g^−1^ ± RSD% DW) in the control and the extract, respectively (see Table 3). In contrast, palmitic, oleic, and stearic acids ranged from 10.41 ± 10.75% to 16.05 ± 8.90%, from 1.22 ± 10.56% to 1.94 ± 6.94 (*p* < 0.05), and from 0.40 ± 16.84% to 0.56 ± 17.47 (mg·g^−1^ ± RSD% DW) in the control and the extract, respectively.

Statistical analyses revealed significant differences in the concentrations (mg·g^−1^) between the extract and the control, with *p* values ranging from <0.05 to <0.01, except for stearic acid, which showed no significant difference (*p* > 0.05).

The total SFAs, PUFAs, and MUFAs were statistically higher in the extracts. The PUFA/SFA rate showed a statistical difference between groups, while the MUFA/SFA and PUFA/MUFA rates did not change (Table 3).

## 4. Discussion

This paper reports on the results of using PBS coupled with UAE to evaluate the feasibility of producing an extract rich in bioactive compounds. Previous research and projects on Spirulina have used common laboratory solvents such as methanol, chloroform, and acetone to extract bioactive compounds, with standard extraction methods such as liquid–liquid and Soxhlet extraction, heat treatment, and maceration. The current trend is to replace solvents with green technologies to fulfil the EU directive on environmentally sustainable production. Therefore, the extraction conditions were selected based on the available literature supporting green solvents as a substitute for standard extraction procedures. Preliminary trials investigated specific solvent extraction rates before setting out the experiment (unpublished data).

The combined extraction method of PBS and UAE was chosen due to the ability of PBS buffer to stabilize proteins and maintain their structural integrity while effectively extracting phycobilin [69,70]. In addition, it facilitates the extraction of water-soluble carbohydrates, particularly those with a low molecular weight or polar sugars; however, a pH of > 6 could limit yields [71].

PBS also solubilizes polar lipids, such as glycerophospholipids (GPLs), and contributes to disrupting protein–lipid complexes. This allows for cell wall destabilization, enhancing membranes and improving the extraction of Chl a and carotenoids [72].

UAE was chosen for its capacity to perform cell disruption in less time and at a lower temperature than other methods, releasing bioactive compounds more efficiently. The combination of PBS and UAE results in a more efficient extraction system. PBS enhances the efficiency of UAE by stabilizing the compound structure (maintaining a stable pH) and increasing acoustic cavitation due to its salt content. This combined effect intensifies cell disruption and improves mass transfer, promoting the release of intracellular compounds from Spirulina [73].

The recent literature confirms the adaptability of Spirulina to various environmental conditions, including temperature, light intensity, and nutrient availability [34,35,36].

The observed biochemical composition of *Arthrospira platensis* (Table 1) aligns with data from the literature, which indicate that its protein content ranges from 55% to 70% dry weight (DW), carbohydrates comprise from 15% to 25% DW, and lipids account for 3% to 9% DW [74,75].

Following the extraction process, the total carbohydrate recovery was 81.16%, lower than the recovery reported by Liu et al. [76]. They utilized an ultrasound-assisted extraction (UAE) pre-treatment at 500 W for 10 min, followed by hot water extraction at 80 °C, achieving a maximum recovery rate of 84.09%. In contrast, the current method employed milder heat conditions and shorter UAE times to minimize the risk of the loss and degradation of bioactive compounds, thereby preserving the nutritional quality of the extract, as reported by Shen et al., who reported that excessive ultrasound power, together with a high frequency and long extraction time, can damage active ingredients [77].

In addition, the findings of this study demonstrated higher recovery rates than those reported by Lupatini et al. [78], who achieved carbohydrate and protein recoveries ranging from 41.52% to 75.56% using more rigorous extraction conditions, including 35 min of sonication at 37 kHz (100% power) followed by 50 min of agitation under alkaline conditions (pH 9) at 30 °C.

The incomplete carbohydrate recovery stated in this study can be attributed to several factors. The cell wall is made of insoluble carbohydrates, such as glucan and peptidoglycan polymers, which form a multilayered structure stabilized by amino acid residues, which cause difficulties in extraction [79]. Additionally, uronic acids and extracellular polymeric substances form gels or aggregates that are more resistant to solubilization than intracellular polysaccharides. Finally, the solubility of glycogen is influenced by factors such as the type of solvent used, extraction temperature, pH, and extraction time [80].

Purdi et al. reported a high protein recovery of 76.83% when using UAE. The recovery shown in this paper (83.35%) confirm UAE’s effectiveness in extracting proteins from the matrix [20]. Studies on pecan and lupin proteins have indicated that higher recoveries can be attributed to the extraction process’s ability to effectively break hydrophobic and hydrogen bonds, thus releasing water-soluble proteins [25,26,27].

The unusually high ash values can be attributed to the salt added during extraction with the phosphate-buffered saline (PBS) solution.

The lipid content in the control was 5.16 g·100 g^−1^ (DW), which aligns with the literature reporting values between 4% and 9% [15,81,82,83]. The complete breakdown of the cell wall resulted in a significantly higher lipid concentration in the extract, with an increase of 20.29% [34,35,36]. This phenomenon is attributed to UAE cavitation, which alters the structure and morphology of plant cell tissue, disrupting or deforming cell walls, making them more porous while breaking down the neutral detergent fiber (NDF) microstructure (including hemicellulose, cellulose, and lignin) [76]. This modification facilitates a higher extraction efficiency and leads to an overall decreased recovery of NDF at 10.48% [77].

Previous works have compared green extraction to standard solvent extractions of the lipid fraction. The yield obtained does not differ consistently among extraction methods; the highest lipid content was obtained with the Bligh and Dyer method (11.6%), followed by high-pressure ethanol (11.4%) [82]. The literature has identified palmitic acid (30–45%), linoleic acid (15–20%), and γ-linolenic acid (GLA) (20–30%) as the primary fatty acids characterizing the lipid profile of Spirulina [78,80,81,82,83]. The data reported in this paper confirm the trend of these constituents, although the absolute amounts cannot be directly compared due to differences in quantification methods.

The increased lipid content in the extracts could account for the higher levels of fatty acids observed, which showed an increase of nearly 60%, consistent with the reported literature [82]. The extraction process has been shown to selectively impact the polyunsaturated fatty acid (PUFA) to saturated fatty acid (SFA) ratio, resulting in a more favorable fatty acid composition [84].

Cyanobacteria synthesize chlorophyll a (Chl a) similarly to higher plants [85]. The Chl a content is influenced by various environmental parameters, including light intensity and nutrient availability, with nitrogen playing a critical role [86]. The reported concentrations of Chl a range from 0.5 to 1.1 mg·100 mg^−1^ DW [87,88], representing approximately 9–12% of the lipid fraction [30].

In the present study, the Chl a concentration in the control samples was measured at 0.49 mg·100 mg^−1^ DW, aligning with Orthesin et al.’s findings [88]. They extracted Chl a using a 90% methanol-based microwave-assisted extraction (MAE) protocol (40 W irradiation for 5 min), resulting in a maximum yield of 0.50 mg·100 mg^−1^ DW. However, the extracts we obtained from UAE showed higher values, reaching 0.64 mg·100 mg⁻^1^ DW (Table 2). In contrast, Park et al. reported lower values for Spirulina (0.26 and 0.47 mg·100 mg^−1^ DW) using acetone extraction with three consecutive 10 min sonication cycles [89]. These findings are consistent with Orthesin’s findings, indicating a lower extraction efficiency for acetone than methanol.

Choi et al. reported nearly three times higher concentrations for *Arthrospira maxima* extracted with a 70% ethanol solution at 80 °C for 24 h (1.38 mg·100 mg^−1^ DW) and with the same solution at 40 °C for 4 h using UAE at 40 kHz (1.80 mg·100 mg^−1^ DW) [90]. The UAE method in their study resulted in a 31% increase in the Chl a concentration, comparable to the 30.61% increase observed in this experiment.

The phycobiliprotein extraction values in this study are consistent with the existing literature, which states that phosphate-buffered saline (PBS) helps to preserve the native structure and stability of phycobiliproteins. UAE has proven effective in extracting these compounds without causing degradation [31,32,33,91]. Chia et al. reported recoveries of −4.9% for phycoerythrin (Pc) and −38% for allophycocyanin (Apc) from Spirulina using UAE (30% amplitude, 5 s ON-5 s OFF, 10 min sonication) combined with liquid biphasic flotation (LBF) (100 cc·min^−1^ air flow rate, 7 min flotation time) [92]. Similarly, Obeid et al. reported a 5% recovery for Pc using UAE (12 kHz and 200 W) and membrane filtration (10 kDa-cutoff membranes) [93]. The results of this study, however, showed higher recoveries of −0.77% for Pc and +41.41% for Apc. Rodrigues et al. used UAE at 25 kHz and 25 °C for 30 min with ionic liquids at pH 6.5, reporting an extraction order of Apc > Pc > Pe; this study observed a different order of Pe > Apc > Pc, likely due to variations in extraction conditions [32].

Da Silva et al. reported a total polyphenol concentration of 2.07 ± 0.01 mg·g^−1^ of gallic acid equivalents (DW) from methanol extracts obtained through UAE technology [94]. In contrast, El-Baky et al. found values ranging from 4.51 to 16.96 mg·g^−1^ gallic acid eq. (DW) when Spirulina was grown in varying salinities and extracted by sonication with ethanol at 4 °C [95]. The extraction method used in this study allowed for the recovery of significant amounts of total polyphenols, which increased by nearly 70% in the extracts.

The DPPH assay showed an antioxidant activity of 21.10 ± 4.58 mg·g^−1^ Trolox eq. DW, consistent with Seghiri et al., who reported an antioxidant activity of 23 mg·g^−1^ Trolox eq. DW in Spirulina methanolic extracts [96]. An enhanced antioxidant activity is usually attributed to a synergistic effect among the components of the antioxidant pool [97,98,99]. In this study, polyphenols increased by almost 66%, phycobiliproteins by 12%, carotenoids by 31%, and Chl a by 32%. The antioxidant properties of Chl a alone are still in the preliminary stages of investigation. In contrast, the antioxidant effects of the other compound families are well known and have been reported in the literature. Polyphenols have a scavenging effect on free radicals and enhance antioxidant enzyme activity [96]. Phycobiliproteins have anti-inflammatory and antioxidant capacities by neutralizing hydroxyl, alkoxyl, and peroxyl radicals, stabilizing cell membranes [100]. Furthermore, carotenoids protect lipid membranes by quenching singlet oxygen, preventing lipid peroxidation, and donating electrons to neutralize radicals [101]. Thus, the increased antioxidant activity observed in this study may be linked to higher concentrations and the synergistic action of polyphenols, phycobiliproteins, and carotenoids in the extracts.

Data on *Arthrospira* spp. in the literature indicates significant levels of group B vitamins, especially vitamin B12 analogues and substitutes, and important amounts of the fat-soluble pro-vitamins A, K, and E [6,10]. However, in this study, only vitamin B3 was detected. Factors such as strain, cultivation system, environmental conditions, and processing techniques can influence the preservation and concentration of vitamins in Spirulina [89]. The lower levels in the extracts may be attributed to the UAE process, which can create localized heat and free radicals that negatively impact the concentrations of heat-sensitive compounds, including vitamins [102].

## 5. Conclusions

The combined use of ultrasonic extraction (UAE) and maceration in phosphate-buffered saline (PBS) has proven effective in extracting bioactive compounds, including phenols, pigments (chlorophyll a, carotenoids, and phycobiliproteins), and essential fatty acids. These compounds significantly enhance the antioxidant properties of extracts. The process has shown satisfactory results in extracting total proteins and carbohydrates, consistent with the existing literature; however, different extraction conditions should be explored to achieve higher yields of proteins and carbohydrates.

The cellulosic cell wall accounts for about 10% of the dry matter of algae, which impacts the accessibility of amino acids and other bioactive compounds, making them indigestible for humans and other non-ruminants. The resulting extracts have a low fiber level, allowing for a higher nutrient bioavailability. Given this composition, the extracts have potential as a food supplement for athletes and bodybuilders, where their high protein content could aid in muscle recovery, especially in low-carbohydrate or ketogenic diets. The potential benefits of the extracts include support for maintaining muscle mass and improving glycemic management in individuals with insulin resistance or type 2 diabetes, due to their favorable ratio of polyunsaturated fatty acids (PUFAs) to saturated fatty acids (SFAs). Natural antioxidants are considered valuable therapeutic agents for reducing illnesses related to oxidative stress. The antioxidant effects of the extracts could positively influence inflammation and play a preventive role in protecting against the generation of free radicals. Studies of the health effects of the extracts alone or loaded into different phospholipid vesicles are ongoing on Caco-cells; even if the data are not conclusive, preliminary results have shown a high biocompatibility and a discrete protective effect against H_2_O_2_-induced oxidative damage to cells at concentrations ranging from 20 to 200 µg·mL^−1^.

The methodology employed is still far from achieving conclusive laboratory-scale results and requires improvements to enhance recovery rates and scale up for industrial applications. This could involve using green energy sources, which would lower processing costs and increase the product’s value.

## Figures and Tables

**Table 1 foods-14-01510-t001:** Biochemical composition (mean g·100 g^−1^ DW) and yield% of control and extract.

Macronutrient	Control	Extract	Yield% Control vs. Extract
	g·100 g^−1^ DW		
Carbohydrate	15.57 ^ay^	12.63 ^b^	−18.84
Lipid	5.16 ^bx^	6.21 ^a^	+20.29
Protein	62.66 ^az^	52.23 ^b^	−16.65
Fiber	3.38 ^az^	0.35 ^b^	−89.52
Ash	8.41 ^bz^	18.55 ^a^	+120.66

Means differences (a, b) between columns were tested using one-way ANOVA with Cochran’s Correction for multiple comparisons. x = *p* < 0.05, y = *p* < 0.01, z = *p* < 0.001.

**Table 2 foods-14-01510-t002:** Photosynthetic pigments (mg·100 mg^−1^), phycobiliproteins, total polyphenols (mg·g^−1^ gallic ac. eq.), and antioxidant activity (mg·g^−1^ Trolox eq.) obtained in control and extract analysis.

Bioactive Compounds	Control	Extract	Yield% Control vs. Extract
Chl a	0.49 ^by^	0.64 ^a^	+32.33
Total carotenoids	0.09 ^by^	0.12 ^a^	+30.84
Pc	10.46 ^a^	10.38 ^a^	−0.77
Apc	3.24 ^bz^	4.58 ^a^	+41.41
Pe	0.66 ^bz^	1.20 ^a^	+81.42
Total polyphenols	5.94 ^bz^	9.86 ^a^	+65.99
Vitamin B3	3.49 ^az^	1.54 ^b^	−55.87
DPPH assay	14.76 ^by^	21.10 ^a^	+42.95

Means differences (a, b) between columns were tested using one-way ANOVA with Cochran’s Correction for multiple comparisons. x = *p* < 0.05, y = *p* < 0.01, z = *p* < 0.001.

**Table 3 foods-14-01510-t003:** Average total fatty acid (FAME) composition content (mg·g^−1^ DW) and yield% of extract vs. control.

FAME	N° C	Control	Extract	Yield% Control vs. Extract
Palmitoleic acid	C 16:1	0.54 ^by^	1.00 ^a^	+88.62
Palmitic acid	C 16:0	10.40 ^bx^	16.05 ^a^	+54.18
γ-linolenic acid (GLA)	C 18:3 (n–6)	1.63 ^by^	2.77 ^a^	+71.15
cis-linoleic acid (LA)	C 18:2 (n–6)	6.90 ^by^	11.5 ^a^	+66.71
cis-Oleic acid	C 18:1 (n–9)	1.23 ^bx^	1.94 ^a^	+58.85
Stearic acid	C 18:0	0.40 ^a^	0.56 ^a^	+40.40
Total FAME (mg·g^−1^)		21.18 ^by^	33.98 ^a^	+60.48
Total PUFA (mg·g^−1^)		1.75 ^by^	2.95 ^a^	+67.55
Total MUFA (mg·g^−1^)		8.52 ^bz^	14.28 ^a^	+67.91
Total SFA (mg·g^−1^)		10.8 ^by^	16.6 ^a^	+53.67
PUFA/SFA		0.79 ^bx^	0.86 ^a^	
MUFA/SFA		0.16 ^a^	0.18 ^a^	
PUFA/MUFA		4.86 ^a^	4.85 ^a^	

Means differences (a, b) between columns were tested using one-way ANOVA with Cochran’s Correction for multiple comparisons. x = *p* < 0.05, y = *p* < 0.01, z = *p* < 0.001.

## Data Availability

The original contributions presented in this study are included in the article. The raw data supporting the conclusions of this article will be made available by the authors upon request.

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
