# Peer review of "Eco-Friendly Extraction to Enhance Antioxidants and Nutritional Value in *Arthrospira platensis"

_foods, 2025, doi:10.3390/foods14091510_

Round 1
Reviewer 1 Report
Comments and Suggestions for Authors
The article is interesting and well written, but there are some details that need to be taken into consideration before publication, namely:
Title should be restructured. It is not correct to say, “extraction of Arthrospira platensis”. What is extracted are compounds from the plant, not the plant. E.g.: Eco-friendly extraction for enhancing antioxidant and nutritional value in Arthrospira platensis. The word “inovative” is a little dubious, since there is already a published paper related to the combination of maceration/ UAE in spirulina (DOI: 10.9755/ejfa.2022.v34.i9.2879)…
Abstract: Please indicate the full names of the abbreviations APC, PE and PC in the abstract. In addition, the abstract does not mention that the study compares spirulina (raw material) vs extract. This is important to emphasize.
The keywords section should be improved.
Introduction:
- Please, write the name of the plant species after the common name.
- It is not necessary to include the abbreviations for SCP and IAAs, because they will not be referred to again in the manuscript.
- “UAE technique, coupled with maceration as pre-treatment, showed increased extraction of bioactive compounds with high antioxidant activity [51-56]”. As it stands, it appears that references 51-56 did UAE extraction coupled with maceration. And that's not what happens. Please rewrite this sentence.
Please review standards and reagents. There are some errors (line 97). In addition, please, group all reagents purchased from the same company into the same sentence (e.g. Sigma). And be consistent in your writing. Some reagents are capitalized, and others are not.
Please, remove periods from subtitles.
Line 119: The abbreviation UAE has already been explained earlier in the introduction. It is not necessary to repeat the full name.
Line 120: Please, replace periods with commas.
Lines 130, 134, 156, 169, 185, 200, 213, 230, 235, 247: It is not correct to start a sentence with a number. Please replace numbers with their respective full names.
Results:
- “Macronutrients” or “macro-nutrients”;“Apc” or “APC”; “Pe” or “PE”; “Pc” or “PC” ? Please, be consistent.
- When the authors compare spirulina and extracts, in fact both are spirulina, but one is spirulina (raw material/powder?) and the other is spirulina (extract). Right? I think it is more correct to say (raw material or powder) vs (extract).
- Line 343: p >0.05 or p<0.05? “x” is missing in Pc in the Table 2.
- Line 346: There is a missing full stop between Table 2 and DPPH.
- Please, delete “± RSD %)” from the results presentation... For instance, the result is 3.49 ± 5.04 mg /100 g.
- Table 2: Pc, Apc and Pe results in the extracts are with comma. Please, replace commas with periods. The same for Table 3.
- Where are the results of the vitamin B group? The text says they are in Table 2, but they are not.
- Where are the results of myristic acid?
- Line 379: Please write γ-linolenic acid in lowercase letters.
Discussion: It is not necessary to include the abbreviation for EPS, because it will not be referred to again in the manuscript.
Author Response
R1 - Title should be restructured.
A1 - The text was revised.
R2 - Many abbreviations appear in the abstract; please give the full name.
A2 - The abstract was implemented as requested (Lines 9-24).
R3 - The keywords section should be improved.
A3 – The keyword section was modified.
R4 - Please, write the name of the plant species after the common name.
A4 – I disagree, sorry.
R5 - It is not necessary to include the abbreviations for SCP and IAAs, because they will not be referred to again in the manuscript.
A5 – The text was corrected.
R6 - “UAE technique, coupled with maceration as pre-treatment, showed increased extraction of bioactive compounds with high antioxidant activity [51-56]”. As it stands, it appears that references 51-56 did UAE extraction coupled with maceration. And that's not what happens. Please rewrite this sentence.
A6 – The references were modified.
R7 - Please review standards and reagents. There are some errors (line 97). In addition, please, group all reagents purchased from the same company into the same sentence (e.g. Sigma). And be consistent in your writing. Some reagents are capitalized, and others are not.
A7 – The section was revised.
R8 - Please, remove periods from subtitles.
A8 – The periods were removed.
R9 - Line 119: The abbreviation UAE has already been explained earlier in the introduction. It is not necessary to repeat the full name.
A9 – The text was corrected.
R10 - Line 120: Please, replace periods with commas.
A10 - The text was corrected.
R11 - Lines 130, 134, 156, 169, 185, 200, 213, 230, 235, 247: It is not correct to start a sentence with a number. Please replace numbers with their respective full names.
A11 – The text was modified.
R12 - “Macronutrients” or “macro-nutrients”;“Apc” or “APC”; “Pe” or “PE”; “Pc” or “PC” ? Please, be consistent.
A12 - The text was modified.
R13 - When the authors compare spirulina and extracts, in fact both are spirulina, but one is spirulina (raw material/powder?) and the other is spirulina (extract). Right? I think it is more correct to say (raw material or powder) vs (extract).
A13 – Yes, you are right. The text was modified.
R14 - Line 343: p >0.05 or p<0.05? “x” is missing in Pc in the Table 2.
A14 – It is p >0.05 as reported in the text (line 338), so the x is not correct in Table 2.
R15 - Line 346: There is a missing full stop between Table 2 and DPPH.
A15 – The text was modified.
R16 - Please, delete “± RSD %)” from the results presentation. For instance, the result is 3.49 ± 5.04 mg/100g.
A16 – The ± RSD % is mandatory to explain if SD or RSD were considered.
R17 - Table 2: Pc, Apc and Pe results in the extracts are with comma. Please, replace commas with periods. The same for Table 3.
A17 - The text was corrected.
R18 - Where are the results of the vitamin B group? The text says they are in Table 2, but they are not.
A18 – The table was corrected
R19 - Where are the results of myristic acid?
A19 – It was an error, myristic acid was not detected.
R20 - Line 379: Please write γ-linolenic acid in lowercase letters.
A20 – The text was corrected.
R21 - Discussion: It is not necessary to include the abbreviation for EPS, because it will not be referred to again in the manuscript.
A21 – The text was corrected.
Reviewer 2 Report
Comments and Suggestions for Authors
This study proposes an ecologically innovative method that effectively improves the extraction efficiency of antioxidants and nutrients from Arthrospira platensis by using ultrasound assisted extraction (UAE) and low-temperature extraction with phosphate buffered saline (PBS). While the methodology is promising and addresses key limitations of traditional extraction techniques, the manuscript requires revisions to strengthen scientific rigor.
- Many abbreviations appear in the abstract, please give the full name.
- Please check whether "Spirulina" needs italics in the manuscript.
- The form of units in the manuscript should be unified, such as "grams" should be changed to "g", "hours" should be changed to "h", please revise the whole text.
- Lines 130 and 177, it is recommended not to start a sentence with a number. Please revise the whole text.
- How are extraction conditions determined?
- The concentrations of PBS are not specified in the manuscript.
- The discussion section lacks depth, and it is suggested to supplement the discussion on mechanisms.
- What are the highlights of this study?
- It is recommended to increase the recovery rates for all components in Table 1.
- The format of the references is not uniform, please modify according to the format requirements of the journal.
Author Response
R1 - Many abbreviations appear in the abstract, please give the full name.
A1 - The text was updated.
R2 - Please check whether "Spirulina" needs italics in the manuscript.
A2 - Spirulina needs italics only when it is associated with platensis. It should be reported with a capital letter; the text was updated.
R3 - The form of units in the manuscript should be unified, such as "grams" should be changed to "g", "hours" should be changed to "h", please revise the whole text.
A3 - The text was revised as requested.
R4 - Lines 130 and 177, it is recommended not to start a sentence with a number. Please revise the whole text.
A4 - The text was revised as requested.
R5 - How are extraction conditions determined?
A5 - The authors revised the literature to select the more efficient extracting conditions. Unpublished trials were done in the laboratory to set specific extraction conditions. The text was implemented to be more explicit (Line 390-394).
R6 - The concentrations of PBS are not specified in the manuscript.
A6 – PBS is a mixture of Na2HPO4, NaH2PO4, and NaCl salts, adjusted to pH 7.4 with HCl 0.1M. PBS does not have proper molarity but is derived from the salt used. The expression usually found in PBS 0.01 M refers to sodium dihydrogen phosphate and sodium hydrogen phosphate, not taking NaCl into consideration. It is more correct to express the single molarity of the constituents. Therefore, the text was corrected accordingly (Lines 109-110).
R7 - The discussion section lacks depth, and the discussion on mechanisms is suggested to be supplemented.
A7 – The text was revised and implemented as requested.
R8 - What are the highlights of this study?
A8 - Highlights
- The study proposed the use of PBS as a green extraction solvent.
- UAE enhanced the extraction of bioactive compounds, improving the antioxidant properties.
- The reduced UAE operating time allows for saving energy and reducing extraction expenses.
R9 - It is recommended that the recovery rates for all components be increased, as shown in Table 1.
A9 – A sentence was added in the text (Line 507-508)
R10 - The format of the references is not uniform; please modify according to the journal's format requirements.
A10 - The references were revised as requested.
Reviewer 3 Report
Comments and Suggestions for Authors
Dear authors,
This investigation presented an Eco-Innovative approach for the extraction of Arthrospira platensis to enhance antioxidants and nutritional value.
The research is not new as a lot of research has already been done with spirulina. The manuscript does present good discussions comparing its results with other research. However, they do not always emphasize the possible effect that the Phosphate buffer may have had on the contents of each extracted component is not always emphasized. Perhaps the novelty is that the authors used the Phosphate buffer as a green solvent, which is one of the positive aspects of this research. Similarly, in this work, reference is made to information that is not worthy and is only mentioned to extend the manuscript. Finally, it seems that the authors did not read the journal guidelines as the format in which they present the manuscript leaves much to be desired.
Other details were noted and are mentioned below.
The manuscript should be reviewed by a native speaker of English, since several errors of interpretation are observed, as well as the incorrect use of semicolons.
Lines17 y 18: What do APC and PE mean?
Line 69: thanks? Use more scientific terminology and less colloquial language.
Line 86: we? Do not personalize.
Lines 90-110: I don't think it is necessary to mention this information.
Lines 117-118: Check if it is the correct citation format.
Lines 130: A sentence should not start with figures. This detail is observed throughout the methodology.
Line 135: plus? Use more scientific terminology and less colloquial language.
Line 198: Specify the meaning of Ch1.
Lines 317-322: I don't think it's necessary to mention every single variable. It is not stated whether the standard deviation or the standard error was considered.
Line 330, 332: Remove the words “Table 1”.
Line 341, 342: I still don’t know what APC or PE mean.
Lines 372-379: Poor writing.
Lines 396-491: The authors make a lot of comparisons with other research. However, they do not always emphasize the possible effect that the Phosphate buffer may have had on the contents of each extracted component.
Line 400: The values reported by Liu are not mentioned.
Tables. The information in the tables is unclear.
Table 1. Poor presentation of the tables. I understand that the literal "a" denotes a higher value than "b", but this is not the case for lipids and ash. Please correct this.
Comments on the Quality of English LanguageThe manuscript should be reviewed by a native speaker of English, since several errors of interpretation are observed, as well as the incorrect use of semicolons.
Author Response
R1 - Lines 17 and 18: What do APC and PE mean?
A1 - The full name was reported as requested (Lines 17-18, 20).
R2 - Line 69: thanks? Use more scientific terminology and less colloquial language.
A2 – The text was changed as requested (Line 69).
R3 - Line 86: we? Do not personalize.
A3 - The text was changed as requested (93).
R4 - Lines 90-110: I don't think it is necessary to mention this information.
A4 – It is the reviewer's opinion; I do not know what his background is. This is an essential part of the Materials and Methods section for chemists in all publications to repeat the experiments correctly. However, the text was revised to fulfill the editor's and other reviewers' requests (Lines 99-120).
R5 - Lines 117-118: Check if it is the correct citation format.
A5 – The text was revised as requested to correct citation format. The correction was also made in another part of the text.
R6 - Lines 130: A sentence should not start with figures. This detail is observed throughout the methodology.
A6 - The text was revised as requested throughout the paper.
R7 - Line 135: plus? Use more scientific terminology and less colloquial language.
A7 – Plus is usually used in scientific terminology. It is not a colloquial word. However, I do not hesitate to change it with a synonym. The text was modified accordingly (Line 146).
R8 - Line 198: Specify the meaning of Ch1.
A8 – It is Chl (a letter, not a number), not Ch1.
R9 - Lines 317-322: I don't think it's necessary to mention every single variable. It is not stated whether the standard deviation or the standard error was considered.
A9 – I agree with the reviewer, the text was modified (Lines 314-321).
R10 - Line 330, 332: Remove the words “Table 1”.
A10 – I do not understand why. Is it redundant for the reviewer?
R11 - Line 341, 342: I still don’t know what APC or PE mean.
A11 – The reviewer can go back to line 212 in the previous version, in the revised version, lines 17-18 and 20, thank you.
Maybe the reviewer means that they are reported in diverse ways, such as APC and Apc, PE and Pe. The text was standardized.
R12 - Lines 372-379: Poor writing.
A12 – The Authors tried to improve the text to be richer (Lines 367-381). Is it poor writing, scientific terminology, or colloquial?
R13 - Lines 396-491: The authors make a lot of comparisons with other research. However, they do not always emphasize the possible effect that the Phosphate buffer may have had on the contents of each extracted component.
A13 – The discussion section was deeply revised. A sentence regarding the effects of PBS and UAE was added (Lines 393-402).
R14 - Line 400: The values reported by Liu are not mentioned.
A14 – The values of Liu (Table 2 in his paper) are reported in Line 409, which is the maximum value of polysaccharide recovery obtained, using the extracting conditions like those reported in our paper.
R15 - Tables. The information in the tables is unclear.
Table 1. Poor presentation of the tables. I understand that the literal "a" denotes a higher value than "b", but this is not the case for lipids and ash. Please correct this.
A15 – The tables were corrected accordingly.
R16 - It seems that the authors did not read the journal guidelines as the format in which they present the manuscript leaves much to be desired.
A16 – The authors followed the “Instructions for Authors” reported in Foods guidelines and the Microsoft Word template. We do not understand what the reviewer is referring to. Some minor discrepancies were adjusted.
R17 - The manuscript should be reviewed by a native speaker of English, since several errors of interpretation are observed, as well as the incorrect use of semicolons.
A17 - We have revised the text for the English wording and sentencing as suggested by the reviewer. We used internationally certified software for grammar correction to verify the correct use of punctuation. However, we find only a few mistakes; moreover, the use of semicolons was correct.
Reviewer 4 Report
Comments and Suggestions for Authors
I reviewed the manuscript entitled Eco-Innovative approach for the extraction of Arthrospira platensis to enhance antioxidants and nutritional value.
I agree to accept this manuscript after major revision.
1) The extraction protocol involved a maceration step in phosphate-buffered saline (PBS, pH 7.4) at 5 °C for 48 hours, followed by ultrasound-assisted extraction (UAE) at 400 W, 30 kHz, 30 cycles/min (with cycles of 1 sec. on/ off, for a total of six extraction cycles). This method substantially improved APC (+41.41%) and PE (+81.42%) yield. The lipid fraction exhibited an increase of 20.29%, including carotenoids (+30.84%), total fatty acids (60.48%), and polyphenols (+65.99%), allowing an increased antioxidant activity (+42.95%). On the contrary, proteins (–16.65%), carbohydrates (-18.84%), and PC (-0.77%) showed decreased recovery rates. 48 hours should change to 48 h. To use international units instead of words, please modify the entire text to address similar issues. APC, PE, PC, etc., using abbreviations directly makes it difficult for readers to understand. The full name should be used for the first time, followed by abbreviations such as ultrasound assisted extraction (UAE). In the abstract, ultrasound assisted extraction (UAE) only appears once, so there is no need to use abbreviations. The full name is sufficient, and abbreviations are only necessary if they appear three or more times, as too many abbreviations can confuse readers.
2) Keywords, Do not use UAE, use ultrasound assisted extraction directly.
3) Spirulina (Cyanophyceae; Arthrospira spp.), Arthrospira is a generic name and should be italicized.
4) enhancing protein digestibility and solubility [23-27,], The comma between 27 and brackets is deleted.
5) UAE has shown high efficiency, reduced solvent consumption, moderate extraction times and costs, and ease of management. Furthermore, it allows faster and more scalable production from the laboratory to the industry [50]. I think the disadvantages of UAE should also be added here, such as high equipment cost, high energy consumption and complex process optimization, because there is no perfect method. Readers should also be made aware of these.
6) D-glucose, α-amylase, D needs italics, and Greek letters also need italics. Check and modify the full text.
7) 0.5 grams of the sample were carbonized at 525 °C in a porcelain crucible for 8 h. Ashes were expressed as g 100 g-1 DW. grams should change to g. g 100 g-1 DW should change to g/100 g DW. Check and modify the full text.
8) 4 ml of a CHCl3/MeOH solution (4:5, v/v) containing tritridecanoin acid (50 mg L-1) as the IS in a 15 ml falcon (F1). ml should change to mL. 0.25μm thickness, There should be spaces between numbers and units. 1162 volts should change to 1162 V.
9) Water-soluble vitamins, It needs to be modified as a title, for example, change it to water-soluble vitamin extraction and analysis.
10) Porspshell 120 EC-C18, 18 requires a subscript.
11) Table 1. The table should be changed to a three line grid; Means differences (a, b), a and b require superscripts. Other tables also have the same problem and need to be modified.
12) whereas PC concentration showed no significant differences (P >0.05), When it comes to statistics, P should be italicized. (p < 0,001), The entire text should be consistent, and it is recommended to use P, 0,001 should change to 0.001.
13) vitamin B group (B1, B3, B2, B6, B9, and B12), changing the numbers of these B vitamins to subscripts is more accurate.
14) There are many instances of 'we' and 'our' in this article, please try to modify them as much as possible, otherwise their excessive appearance may make people feel that the viewpoint of this study is not objective enough.
15) The fatty acid profile of spirulina is mainly characterized by palmitic (45%), linoleic, and γ-linolenic acid (GLA) (20%), what is the proportion of linoleic acid? Please add it.
16) The higher lipid amount in the extracts could explain the higher values of fatty acids, which showed an increase of almost 60%, with values consistent with literature data. Please add relevant literature.
17) According to Table 1, the fiber content decreased by 89.52%. Has the author considered the impact of this significant decrease on the final product? Will it affect its fiber content as a dietary supplement?
18) Table 2 shows that the content of carotenoids and chlorophyll-a increased by 30.84% and 32.33%, respectively. Has the author considered the impact of these additions on the color and nutritional value of the final product?
19) The author mentioned the use of PBS buffer and ultrasound assisted extraction (UAE), but did not discuss in detail why these conditions were chosen. Have you considered comparing other solvents or extraction methods?
20) The author mentioned “The enhancement in antioxidant activity in our samples can be attributed to the increased concentrations of polyphenols, phycobiliproteins, and carotenoids in the extract.” But there was no detailed discussion on the specific mechanism of this synergistic effect. Can you provide more explanations on the synergistic effects between polyphenols, phycobiliproteins, and carotenoids?
21) The author did not discuss the safety issues of the extract. Has any relevant toxicology or safety assessment been conducted?
22) Conclusions, This section should not only include the focus of the research, but also the application, future prospects, and limitations of the study.
23) This study developed an eco-innovative method to enhance the antioxidant and nutritional values of Spirulina (Arthrospira platensis), a microalgae with a global production of ~18,000 tons/year, widely used in dietary supplements. The extraction protocol involved maceration in PBS (pH 7.4) at 5°C for 48 hours, followed by ultrasound-assisted extraction at 400 W, 30 kHz, 30 cycles/min (1 sec. on/off, six cycles). This method significantly improved APC (+41.41%) and PE (+81.42%) yields. The lipid fraction increased by 20.29%, including carotenoids (+30.84%), total fatty acids (+60.48%), and polyphenols (+65.99%), enhancing antioxidant activity (+42.95%). However, proteins (–16.65%), carbohydrates (–18.84%), and PC (–0.77%) showed reduced recovery. This approach provides a promising method for extracting high-value compounds from Spirulina, supporting its potential applications in the dietary supplement sector.
24) I have read all the references and found some issues. Ref 4, delete the contents of the issue; Ref 5, pp. should be deleted. The format of DOI should be the same as Ref 4. Ref 26, The year after the author's name should be deleted because it already exists below. Ref 33, 603-607.29. should change to 603-607. Ref 54, the volume does not have italics. I have just listed some examples, please carefully review and revise them. The author also needs to check and revise according to the requirements of the journal.
25) The biggest problem with this study is many details need to be modified and improved. Too many issues can make people feel that the author's attitude is not rigorous.The author must take them seriously and make necessary revisions.
26) Language needs polishing.
Comments on the Quality of English LanguageLanguage needs polishing.
Author Response
R1 - 48 hours should change to 48 h. To use international units instead of words, please modify the entire text to address similar issues.
A1 – The entire text was corrected as requested.
R2 - APC, PE, PC, etc., using abbreviations directly makes it difficult for readers to understand. The full name should be used for the first time, followed by abbreviations such as ultrasound assisted extraction (UAE). In the abstract, ultrasound assisted extraction (UAE) only appears once, so there is no need to use abbreviations. The full name is sufficient, and abbreviations are only necessary if they appear three or more times, as too many abbreviations can confuse readers.
A2 – From the instructions for Authors:
Acronyms/Abbreviations/Initialisms should be defined the first time they appear in each of three sections: the abstract, the main text, and the first figure or table.
When defined for the first time, the acronym/abbreviation/initialism should be added in parentheses after the written-out form. After the first time, only the abbreviation should be inserted in the text.
We revised the text to accomplish the instructions.
R3 - Keywords, Do not use UAE, use ultrasound assisted extraction directly.
A3 – The text was corrected as requested.
R4 - Spirulina (Cyanophyceae; Arthrospira spp.), Arthrospira is a generic name and should be italicized.
A4 - The text was corrected as requested.
R5 - enhancing protein digestibility and solubility [23-27,], The comma between 27 and brackets is deleted.
A5 – The text was corrected as requested.
R6 - UAE has shown high efficiency, reduced solvent consumption, moderate extraction times and costs, and ease of management. Furthermore, it allows faster and more scalable production from the laboratory to the industry [50]. I think the disadvantages of UAE should also be added here, such as high equipment cost, high energy consumption and complex process optimization, because there is no perfect method. Readers should also be made aware of these.
A6 – A sentence was added for completeness (Lines 79-86).
R7 - D-glucose, α-amylase, D needs italics, and Greek letters also need italics. Check and modify the full text.
A7 – Sorry, there is no standard rule; In my expertise, D does not need italics, nor Greek letters. I left the decision to the editor.
R8 - 0.5 grams of the sample were carbonized at 525 °C in a porcelain crucible for 8 h. Ashes were expressed as g 100 g-1 DW. grams should change to g. g 100 g-1 DW should change to g/100 g DW. Check and modify the full text.
A8 – Quantity expression was set following SI guidelines.
R9 - 4 ml of a CHCl3/MeOH solution (4:5, v/v) containing tritridecanoin acid (50 mg L-1) as the IS in a 15 ml falcon (F1). ml should change to mL. 0.25 μm thickness, There should be spaces between numbers and units. 1162 volts should change to 1162 V.
A9 – The entire text was corrected as requested.
R10 - Water-soluble vitamins, It needs to be modified as a title, for example, change it to water-soluble vitamin extraction and analysis.
A10 – The headings of the paragraph are all written the same way. We do not believe it is appropriate to change only the title of the vitamins, leaving the others as they are.
R11 - Porspshell 120 EC-C18, 18 requires a subscript.
A11 - Sorry, but this is the right way to write the stationary Phase of an analytical column.
R12 - Table 1. The table should be changed to a three line grid; Means differences (a, b), a and b require superscripts. Other tables also have the same problem and need to be modified.
A12 – We believe that the structure of the tables is correct. The mean difference’s locations were changed in superscript as indicated in the instructions for the authors.
R13 - whereas PC concentration showed no significant differences (P >0.05), When it comes to statistics, P should be italicized. (p < 0,001), The entire text should be consistent, and it is recommended to use P, 0,001 should change to 0.001.
A13 – The text was corrected as necessary.
R14 - vitamin B group (B1, B3, B2, B6, B9, and B12), changing the numbers of these B vitamins to subscripts is more accurate.
A14 – I disagree, it is not a problem of accuracy, it is only related to subjective preferences.
R15 - There are many instances of 'we' and 'our' in this article, please try to modify them as much as possible, otherwise their excessive appearance may make people feel that the viewpoint of this study is not objective enough.
A15 – The text was changed as requested.
R16 – Line 431-434. The fatty acid profile of spirulina is mainly characterized by palmitic (45%), linoleic, and γ-linolenic acid (GLA) (20%), what is the proportion of linoleic acid? Please add it.
A16 – The sentence was modified (Lines 443-450).
R17 - The higher lipid amount in the extracts could explain the higher values of fatty acids, which showed an increase of almost 60%, with values consistent with literature data. Please add relevant literature.
A17 – The reference was added.
R18 - According to Table 1, the fiber content decreased by 89.52%. Has the author considered the impact of this significant decrease on the final product? Will it affect its fiber content as a dietary supplement?
A18 – As reported in the text (Lines 51-54), the relatively low algae amino acid digestibility has been related to high levels of fibers.
Therefore, this type of extract with low fiber content is especially indicated for dietary supplements with high protein and antioxidant content.
In the conclusion section, a sentence was added in the text (Lines 521-523).
R19 - Table 2 shows that the content of carotenoids and chlorophyll-a increased by 30.84% and 32.33%, respectively. Has the author considered the impact of these additions on the color and nutritional value of the final product?
A19 – The impact on the color was not considered, whereas the impact on the nutritional value was highlighted (Lines 497-504)
R20 - The author mentioned the use of PBS buffer and ultrasound assisted extraction (UAE), but did not discuss in detail why these conditions were chosen. Have you considered comparing other solvents or extraction methods?
A20 – The text was improved (Lines 393-407)
R21 - The author mentioned “The enhancement in antioxidant activity in our samples can be attributed to the increased concentrations of polyphenols, phycobiliproteins, and carotenoids in the extract.” But there was no detailed discussion on the specific mechanism of this synergistic effect. Can you provide more explanations on the synergistic effects between polyphenols, phycobiliproteins, and carotenoids?
A21 – A sentence was added for completeness (Lines 504-514).
R22 - The author did not discuss the safety issues of the extract. Has any relevant toxicology or safety assessment been conducted?
A22 – A sentence was added for completeness (Lines 546-550).
R23 – Conclusions. This section should not only include the focus of the research, but also the application, future prospects, and limitations of the study.
A23 – The conclusions section was revised.
R24 - I have read all the references and found some issues. Ref 4, delete the contents of the issue; Ref 5, pp. should be deleted. The format of DOI should be the same as Ref 4. Ref 26, The year after the author's name should be deleted because it already exists below. Ref 33, 603-607.29. should change to 603-607. Ref 54, the volume does not have italics. I have just listed some examples, please carefully review and revise them. The author also needs to check and revise according to the requirements of the journal.
A24 – Thank you, the reference section was deeply revised.
R25 - The biggest problem with this study is many details need to be modified and improved. Too many issues can make people feel that the author's attitude is not rigorous.The author must take them seriously and make necessary revisions.
A25 – The text was deeply revised to resolve the gaps detected by the reviewer.
R26 - Language needs polishing.
A26 – The text was revised for English.
Round 2
Reviewer 2 Report
Comments and Suggestions for Authors
I think authors gave a pretty good response to all questions. The present manuscript could be acceptable for publication.
Author Response
Thank you
Reviewer 3 Report
Comments and Suggestions for Authors
Dear authors,
This investigation presented an Eco-Innovative approach for the extraction of Arthrospira platensis to enhance antioxidant and nutritional value.
The manuscript has been moderately improved. However, there are still issues that were highlighted in the previous review which haven’t been dealt with by the authors.
Comments on the Quality of English LanguageThe English could be improved to more clearly express the research.
Author Response
Dear Reviewer
17 suggestions/comments were reported.
Among them, the comments R1, R2, R3, R5, R6, R7, R8, R9, R12, R13, R14, and R15 were corrected/improved as requested.
Regarding R2, the word thanks was changed in line 532
R4 - Lines 99-120. The paragraph Standards and reagents is necessary for the Materials and Methods Chapter.
R10 - I do not believe that Table 1 should be deleted. If the Editor agrees with the reviewer, we will delete it.
R11 - I think we answered the reviewer's request.
R13 - The text was implemented. Line 394-408.
R16 - The text has been checked for compliance with the guidelines.
R17 - The text was revised for English wording by an English colleague who teaches in the course of English for our students. Moreover, an internationally certified software for grammar correction was used to verify discrepancies in using semicolons.
To meet the reviewer's request, we tried to improve the text in:
R18 Is the research design appropriate? Must be improved.
To improve the research design, new trials and analyses should be performed; I do not think it could be done in the present paper. Studies on bioavailability and antioxidant capacity on cellular lines are ongoing, but data are not conclusive and are reported in the present paper.
R19 Are the results clearly presented? Must be improved.
The text was revised, but we do not see how it could be implemented as the reviewer requested.
In the discussion, the text was modified (Lines 394-408)
R20 Are the conclusions supported by the results? Can be improved.
Lines 531-534, the text was modified.
Reviewer 4 Report
Comments and Suggestions for Authors
The author has made modifications and explanations according to my requirements, and I agree to accept the work in its current form.
Author Response
Thank you